# Frailty, Health Literacy, and Self-Care in Patients with Chronic Kidney Disease in Taiwan

**DOI:** 10.3390/ijerph19095350

**Published:** 2022-04-28

**Authors:** Mu-Dan Tsai, Jen-Pi Tsai, Min-Li Chen, Li-Chun Chang

**Affiliations:** 1Department of Nursing, Douliou Tzu Chi Hospital, Yunlin City 64041, Taiwan; muda560@tzuchi.com.tw; 2Department of Nephrology, Dalin Tzu Chi Hospital, Chiayi City 62247, Taiwan; dm315797@tzuchi.com.tw; 3School of Medicine, Tzu Chi University, Hualien 97004, Taiwan; 4School of Nursing, Chang Gung University of Science and Technology, Chiayi City 61363, Taiwan; mlchen@gw.cgust.edu.tw; 5School of Nursing, Chang Gung University of Science and Technology, Taoyuan City 33303, Taiwan; 6Chang Gung Memorial Hospital, Keelung 20401, Taiwan; 7Department of Nursing, Chang Gung University, Taoyuan City 33302, Taiwan

**Keywords:** frailty, health literacy (HL), chronic kidney disease (CKD), self-care

## Abstract

Chronic kidney disease (CKD) is a chronic and often irreversible disease that requires active self-care to mitigate adverse outcomes. This study aimed to analyze the associations of demographic and disease data, frailty, health literacy (HL), and CKD self-care (CKDSC) in patients with CKD. We conducted a cross-sectional study at two hospitals in Taiwan. A total of 144 CKD patients with a mean age of 66.8 ± 9.1 years were included in the study. Among them, 79.2% were in CKD G3, and the mean time since diagnosis of CKD was 86 ± 48 months. Approximately 62.5% were identified as non-frail. The mean of HL and CKDSC were 11.76 ± 4.10 and 62.12 ± 9.31. In multivariate linear regression analysis, age ≥ 65 years (odds ratio (OR) = 5.67, 95% confidence interval (CI) 1.59–9.75), non-frailty (OR = 2.19, 95% CI 0.02–5.40), and high critical HL (OR = 1.43, 95% CI 0.13–2.90) showed significant positive correlation with CKDSC. Therefore, management of patients with CKD should focus on the young population, reinforcing health education strategies that improve critical HL and preventing frailty that may interfere with self-care. In addition, the patient’s social support resources should be expanded to achieve the goal of CKDSC.

## 1. Introduction

Chronic kidney disease (CKD) is a serious global health problem, and patients require active self-care [1]. The United States Renal Data System reported in 2016 that Taiwan has the largest incidence and prevalence of renal failure [2]. CKD is defined as estimated glomerular filtration rate (eGFR) < 60 mL/min/1.73 m^2^ for more than 3 months and the five stages classified by eGFR were stage 1, eGFR ≥ 90 mL/min/1.73 m^2^; stage 2, 60–89 mL/min/1.73 m^2^; stage 3a, 45–59 mL/min/1.73 m^2^; stage 3b, 30–44 mL/min/1.73 m^2^; stage 4, 15–29 mL/min/1.73 m^2^; and stage 5,< 15 mL/min/1.73 m^2^ [3]. Moreover, the Kidney Disease Improving Global Outcomes (KDIGO) organization has summarized the stages of CKD with the classification of five levels of dysfunction defined by eGFR (G1–G5) and three by albuminuria (A1–A3) [4,5].

A recent study in Taiwan showed that the overall prevalence of CKD stages 1 to 5 was 15.5% and that of CKD stages 3 to 5 was 9.1%, with an incidence of nearly 27.2 per 1000 people per year [6]. According to Taiwan’s National Health Insurance Statistics in 2018, the highest medical expenditure of the National Health Insurance was attributed to CKD. The cost covered for dialysis from the National Health Insurance outpatient services was estimated to be NT $56.2 billion, accounting for 9.2% of the overall National Health Insurance budget. Thus, CKD affects population health but is also a serious financial burden for the national medical resources [7].

The prevalence of CKD increases with age, especially in older adults (>65 years) [8]. Frailty is described as an age-related clinical state that can be a predictive factor for falls, disability, hospitalization, and death of the elderly [9]. The evidence that the prevalence of frailty in patients with CKD is higher than that in the general population needs to be evaluated [10,11]. Frailty is an important issue in patients with CKD, with prevalence reported between 25% and 80%, which is likely a significant contributing factor to multiple adverse health events [12]. CKD patients with frailty have impeded daily activities and increased healthcare utilization [11]. When psychomotor speed is slightly impaired, it can negatively impact patient engagement in CKD self-care (CKDSC) [13].

Self-care is an essential component in the long-term management of CKD. It is defined as the process of maintaining health through health-promoting practices and illness management [14]. The core strategy in CKD case management provided by nurses is to increase CKDSC behaviors [15]. Previous studies have demonstrated that CKD patients with better self-care behavior have a lower risk of developing a rapid decline in renal function [16,17]. Sex, age, stage of CKD, and comorbidities were also significant factors for CKDSC [14,17,18]. Recently, health literacy (HL) has been demonstrated to have a significant role in promoting self-care in patients with CKD [19,20]. For elderly patients, exertional fatigue or weakness can lead to physical impairment and depression [21] and decrease self-care capabilities [22,23]. However, the correlation between frailty, HL, and self-care in patients with CKD has not been well explored. The aim of this study was to investigate the relationship between frailty and HL, including accessing, understanding, appraising, and applying health information, communication/interaction, and self-care behavior in patients with CKD.

## 2. Materials and Methods

Patients with CKD G3 to G5 who were not undergoing dialysis were recruited from two hospitals in southern Taiwan, using a cross-sectional study design and a convenience sampling method. The institutional review board at Buddhist Dalin Tzu Chi Hospital approved all study procedures (no B10804005), and written consent was obtained from all study participants.

### 2.1. Participants

Patients were recruited and interviewed in an outpatient clinic between January and March 2020. The inclusion criteria were as follows: (1) diagnosed with CKD G3 to G5 by a nephrologist at least 1 year prior, with continued nephrology visits; (2) aged 40 years or older; and (3) conscious and able to communicate clearly and speak in Mandarin or Taiwanese dialect. The exclusion criteria were as follows: (1) having a cognitive impairment or mental illness; (2) receiving kidney replacement therapy or kidney transplant; and (3) unable to perform daily activities without assistance.

The suggested sample size was calculated using G power analysis with the power set at 0.80, and the medium effect size at 0.50 was 131 patients [24]. Therefore, a total of 158 CKD patients were approached, and 144 valid questionnaires were completed (response rate, 91%).

### 2.2. Measurement

Demographics: Data included sex, age, educational qualification, and marital status. Disease characteristics—CKD stage, time since CKD diagnosis, and comorbidities (diabetes mellitus, hypertension, hyperlipidemia, and heart disease)—were recorded from hospital records.Frailty: The easy-to-apply study of osteoporotic fractures (SOF) criteria developed by Ensrud et al. [25] for community-dwelling older outpatients was used. Frailty was identified by the presence of at least two of the following three components, and pre-frailty was defined by the presence of one of the following three components: (1) weight loss of 5% or more in the last 2–3 years, irrespective of the intent to lose weight; (2) the subject’s inability to rise from a chair five times without using the arms; and (3) reduced energy level, as identified by an answer of “no” to the question “Do you feel full of energy?” on the geriatric depression scale.CKD-specific HL: The CKD-specific HL in Mandarin and Taiwanese used in our study was developed by Wei et al. [20]. The process of development was based on patient input, panel discussions with experts, and a literature review, and checked for validity and reliability in a pilot test. Moreover, the factorial structure of the items was tested by confirmatory factor analysis, leaving 17 items to measure HL consisting of functional literacy (5 items), communicative literacy (7 items), and critical literacy (5 items). Finally, the 12 items were presented as a multiple-choice cloze test, with one point awarded for each correct response, presenting excellent reliability and validity.CKDSC scale: We adopted the CKDSC as a 16-item questionnaire with 5 subscales: medication adherence (5 items), diet control (4 items), exercise (3 items), smoking behaviors (2 items), and blood pressure monitoring (2 items). Based on the Likert scale, responses ranged from 1 (almost never) to 5 (almost always). Five items in the medication adherence subscale were negatively worded and needed to be reverse-recoded [26]. The CKDSC total score was the sum of each score from the 16 items and ranged from 16 to 80 points. Higher scores indicated better self-care behaviors. The content validity index and Cronbach‘s alpha of the original scale were 0.97 and 0.83, respectively [19].

### 2.3. Data Collection

CKD-specific HL and CKDSC questionnaires were permitted by the authors for use in this study. Data were collected through face-to-face interviews and medical record reviews from January to March 2020. A structured questionnaire was used to collect the data. All subjects participated in the present study voluntarily and could withdraw from the study at any point without penalty. All completed questionnaires were assigned numbers to be identified but remained anonymous during surveys, categorizing, and data analysis.

### 2.4. Statistical Analysis

Frequencies were used to present the proportions of demographic and clinical characteristics, frailty, HL, and self-care among the CKD patients. In addition, bivariate and multivariate linear regression analyses were used to explore factors associated with CKDSC scores. Multivariate models included all variables listed in Table 1 with a *p*-value < 0.05 in the univariate analysis of CKDSC. All statistical analyses were performed using SPSS version 20 (IBM Corp., Armonk, NY, USA). The level of significance was set at *p* < 0.05.

## 3. Results

### 3.1. Descriptive Statistics

The demographic and disease characteristics of the 144 participants are shown in Table 1. The mean age of the participants was 66.8 ± 9.1 years. Among the 144 participants, 105 (72.29%) were male, 48 (75.7%) patients were married, 84 (58.3%) had completed senior high school or above, and 93 (64.6%) reported a monthly family income of ≤ 700 USD. Based on the stage of CKD, 79.2% were in G3, and the mean time since diagnosis was 86 months (7.2 years) (SD = 48). Further, in terms of comorbidities, 90.3% reported having diabetes and 45.1% had at least three comorbidities. Of the total participants, 62.5% were identified as non-frail.

Table 2 shows that the mean HL score was 11.76 ± 4.10, and the scores for functional, communicative, and critical HL were 3.44 ± 1.69, 4.78 ± 1.50, and 3.54 ± 1.43, respectively. The mean self-care behavior score was 62.12 ± 9.31. The highest score of all subscales was medication adherence (Mean = 4.8 ± 0.6) such as in the item “I myself may change prescribed drug dosage (Mean 4.9 ± 0.5)”(the reverse item), and the lowest one was blood pressure monitoring subscale (Mean 3.1 ± 1.6) such as in the item “I always monitor my blood pressure (Mean 2.9 ± 1.6)” (see Appendix A).

### 3.2. The Associations between Demographic Data, Clinical Characteristics, Frailty, Health Literacy, and CKDSC

All variables were examined using bivariate analysis linear regression analysis to identify the determinants of self-care behavior (Table 3). Self-care behavior significantly correlated with age (OR = 4.95, 95% CI 1.93–7.96), single marital status (OR = −3.66, 95% CI −7.20–−0.13), CKD stages 4 and 5 (OR = 2.67, 95% CI 0.16–5.18), non-frailty (OR = 2.21, 95% CI 0.14–4.28), and critical health literacy (OR = 1.37, 95% CI 0.57–3.17). In multivariate analysis, age ≥ 65 years (OR = 5.67, 95% CI 1.59–9.75), non-frailty (OR = 2.19, 95% CI 0.02–5.40), and HL (OR = 1.43, 95% CI 0.13–2.90) were found to be independent factors for CKDSC.

## 4. Discussion

This is the first study to investigate the association of frailty and different dimensions of HL with the self-care behavior of patients with CKD. The results showed a significant correlation between the three variables. Most of the subjects in this study were patients with CKD stages 3 and 4. It was found that factors affecting self-care behavior in these patients included age, disease course, frailty, and critical HL. These findings can serve as points of reference for the design of care strategies for patients with CKD.

This study recruited patients from rural cities in southern Taiwan. The mean age, education level, marital status, disease history, and mean scores of self-care were consistent with those listed in previous studies [14,21]. Similar to the CKD study by Tsai et al. [18], self-care negatively correlated with age implying that the elderly had better self-care behavior. Awareness of CKD is necessary for patient engagement and adherence to medical regimens [27]. The awareness of poor physical function and multiple morbidities might influence elderly patients to have better self-care behavior [18] than younger patients who may be in denial because of the lack of symptoms during the early stages of CKD [17]. Thus, low-to-moderate exercise intensity and duration of training may induce favorable improvements in functional and physical performance markers in CKD patients [28]. Additionally, accumulating evidence shows that younger age is correlated with poor self-care behavior in chronic diseases, such as diabetes [29] and heart failure [30]. Thus, health professionals must prioritize improving the self-care behavior of young patients with CKD when designing self-care programs.

In this study, 37.5% of the participants were diagnosed with frailty. In previous studies performed on patients with CKD, the prevalence of frailty ranged from 16% to 88% for elderly adults, depending on the method and the definition adopted to identify frailty [12,31,32]. In the present study, the proportion of patients with frailty was higher than that of a previous study that adopted a similar approach to diagnose frailty in 693 elderly patients in Korea [33] and lower than that reported by Bilotta et al. [34] in 265 community-dwelling outpatients aged > 65. Moreover, frailty prevalence in the CKD population can impede the patient’s physical function with low grip strength, gait speed, and muscle mass [8]. Therefore, patients with CKD who did not suffer from frailty experienced less mobility restriction and consequently, better self-care ability. To determine the influence of frailty on CKDSC, further research will be necessary to examine the relationship between social support, frailty, and self-care in patients with CKD.

The mean scores of the three domains of CKD-specific HL in our study were higher than those reported in the study conducted by Wei et al. [20] in 1155 patients with CKD with a mean age of 67.48 years from 10 medical centers, 18 regional hospitals, and 15 local hospitals in Taiwan. We recruited CKD patients from outpatient departments in medical centers that conducted the early CKD care program supported by Taiwan’s Ministry of Health and Welfare, thereby improving the educational aspects of HL. Previous studies reported that HL could influence self-care behaviors in patients with diabetes; in particular, critical HL played a greater role in successful self-management than functional HL and communicative HL [35,36]. HL is also crucial to ensure that patients with CKD understand their illness and make decisions based on this knowledge [20,37]. In a previous study by Chollou et al. [38] conducted on 192 patients with diabetes having a low level of education, approximately 28.8% of the variation in self-care behaviors was explained by the HL and demographic variables.

This study has several limitations. First, HL and self-care behavior were evaluated using a cross-sectional design; therefore, further prospective studies are needed to evaluate the dynamic impact of HL on self-care behavior. Second, we used questionnaires to evaluate health literacy and self-care behavior, and therefore, recall bias may have affected the results. Third, the sampling settings were outpatient departments in the hospital, and the majority of the participants had CKD G 3 or G4, which limited the ability to generalize the study findings to other populations with CKD. To overcome the limitation of a small sample size involving young patients with CKD, we recommend a longitudinal study design to understand if CKDSC is a persistent issue affecting patients across the disease spectrum. Further, critical HL can affect the quality of CKDSC. Thus, health education should strengthen the critical CKDSC information and reinforce the implementation of self-care strategies tailored for various life situations. In addition, clinical care of CKD patients should identify those with frailty, and health education should focus on resolving the restrictions of self-care due to frailty and improving the quality of self-care by alleviating frailty. In this study, frailty was assessed using SOF criteria. As a simple and efficient approach, SOF was adopted in fall assessment of the elderly in the community; however; its use among patients with CKD was unprecedented. Therefore, future studies should incorporate other frailty assessments to facilitate a comparison with the self-care of patients with other chronic diseases.

## 5. Conclusions

This study found that patients with CKD demonstrated a moderate degree of HL, and 37.5% were diagnosed with frailty or pre-frailty. Patients who were older than 65 years, not frail, and had a high critical health literacy showed better CKDSC behavior. Thus, CKD case management should be designed to individualize health education for the elderly and those with frailty to strengthen critical HL and improve self-care.

## Figures and Tables

**Table 1 ijerph-19-05350-t001:** Demographic and disease data of CKD patients (N = 144).

	N	%
Sex		
Male	105	72.9
Female	39	27.1
Age (in years)	Mean = 66.8 ± 9.1	Range 44~87
<65	56	38.9
≥65	88	61.1
Highest level of education		
Junior high school and below	60	41.7
Senior high school and above	84	58.3
Marriage status		
Married/partnership	109	75.7
Single	35	24.3
Family income(month) USD		
none	32	22.2
<700	61	42.4
701–1500	30	20.8
>1501	21	14.6
Stages		
G3	114	79.2
G4	27	18.8
G5	3	2.1
Duration of CKD diagnosis in months	Mean = 86 ± 48	Range 6–240
≤84	82	56.9
>84	62	43.1
Number of comorbid conditions	Mean = 2.8 ± 0.9	Range 0–4
0	1	0.7
1	8	5.6
2	38	26.4
3	65	45.1
≥4	32	22.2
Frailty		
No	90	62.5
Pre-frailty/frailty	54	37.5

CKD: chronic kidney disease; USD: United States dollar.

**Table 2 ijerph-19-05350-t002:** Descriptive results of subscales of health literacy and CKD self-care.

Variables	Mean	SD	Range
Health literacy	11.76	4.10	0–17
Functional health literacy	3.44	1.69	0–5
Communicative health literacy	4.78	1.50	0–7
Critical health literacy	3.54	1.43	0–5
Chronic kidney disease self-care	62.12	9.31	36–78

SD: standard deviation.

**Table 3 ijerph-19-05350-t003:** Multiple linear regressions of CKDSC.

Variables	Bivariate		Multivariate	
	OR	(95% CI)	OR	(95% CI)
Female	1.81	−1.65–5.26	-	
Age ≥ 65	**4.95**	**1.93–7.96**	**5.67**	**1.59–9.75**
Junior high school and above	0.35	−2.78–3.47	-	-
Single	**−3.66**	**−7.20–−0.13**	2.25	−5.82–1.32
Monthly income > USD 700	0.40	−2.14–6.54	-	-
CKD G 4 and 5	**2.67**	**0.16–5.18**	2.92	−0.66–6.49
CKD diagnosis > 84 months	4.07	−1.03–7.11	-	-
Having > 3 comorbid conditions	−0.07	−1.83–1.67	-	-
Non-Frailty	**2.21**	**0.14–4.28**	**2.19**	**0.02–5.40**
Functional health literacy	0.60	−0.75–1.94	-	-
Communicative health literacy	0.54	−3.20–4.09	-	-
Critical health literacy	**1.37**	**0.57–3.17**	**1.43**	**0.13–2.90**

R^2^ = 0.206; CKDSC: CKD self-care; OR: odds ratio; CI: confidence interval; bold: *p* < 0.05.

## Data Availability

The data presented in this study are available on request from the corresponding author.

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
