# Peer review of "Frailty, Health Literacy, and Self-Care in Patients with Chronic Kidney Disease in Taiwan"

_ijerph, 2022, doi:10.3390/ijerph19095350_

Round 1

Reviewer 1 Report

Authors present a cross-sectional study evaluating in 144 CKD pts links between frailty and self-care. No major notes.

The only suggestions are:

1) to modify reporting in Tables 1,2 and 3 using an alignment for text on the left for the first column, so as to facilitate reading of relative results.

2) In 1st page at line 37 reference 4 is precedeed by reference 5 on line 36; the sequence of these 2 references should be reordered.

Author Response

Authors responses to decision letter

We are grateful for the positive and constructive comments that originated from the review process. We have responded to the points raised by the Editor and reviewers as follows, with new material highlighted in red text.

Reviewer 1#

The only suggestions are:

1) to modify reporting in Tables 1,2 and 3 using an alignment for text on the left for the first column, so as to facilitate reading of relative results.

2) In 1st page at line 37 reference 4 is precedeed by reference 5 on line 36; the sequence of these 2 references should be reordered.

Response to reviewer

  1. Table 1, 2 and 3 have revised according the reviewer’s suggestion, please see the page 4, line 150-168.
  2. We have rechecked the reference number and recorded it.

Reviewer 2 Report

This study was designed to investigate the relationships between frailty and, health literacy (HL), including accessing, understanding, appraising, and applying health information, communication/interaction, and self-care behavior in patients with CKD. Authors found that management of patients with CKD should focus on young populations, reinforce health education strategies improving critical HL, and incorporate the fact that frailty may interfere with self-care. This manuscript gives new information, but several issues should be resolved to improve the manuscript.

  1. Authors adopted the CKDSC as a 16-item questionnaire with five subscales: medication adherence (five items), diet control (four items), exercise (three items), smoking behaviors (two items), and blood pressure monitoring (two items). Based on the Likert scale, responses ranged from 1 (almost never) to 5(almost always). Five items in the medication adherence subscale were negatively worded and needed to be reverse-recoded. Please explain and describe each item and make new tables for results of item as supplementary data.
  2. Authors included patients aged 40 years or older. However, authors described the prevalence of CKD increases with age, especially in older adults (> 65 years) in the introduction part and titled as “Frailty, Health Literacy, and Self-Care in Elderly Adults with Chronic Kidney Disease in Taiwan”. Therefore, excluding patients aged < 65 years is more adequate for this manuscript. In addition, only 3 CKD 5 patients were enrolled in this study. Therefore, it is better to analyze data excluding CKD 5 patients.
  3. Authors concluded that younger age is correlated with poor self-care behavior and patients who were older than 65 years, showed better CKDSC behavior. Please add the mean age and CKDSC in patients aged > 65 years and aged < 65 years in the table 1. In addition, authors should define the younger age. Please additionally analyze linear regression of CKDSC with age not age ≥ 65 years or not. Authors also concluded that CKD case management should design individualized health education for the elderly and those with frailty, strengthen critical HL, and improve self-care. Please precisely discuss them how to design.
  4. CKD-specific HL: The CKD-specific HL in Mandarin and Taiwanese was developed based on patient input, panel discussions with experts, and a literature review, and checked for validity and reliability in a pilot test. The factorial structure of the items was tested by confirmatory factor analysis, leaving 17 items to measure HL consisting of functional literacy (five items), communicative literacy (seven items), and critical literacy (five items). The 12 items were presented as a multiple-choice cloze test, with one point awarded for each correct response. Please explain and describe each item as supplementary data.

Author Response

Authors responses to decision letter

We are grateful for the positive and constructive comments that originated from the review process. We have responded to the points raised by the Editor and reviewers as follows, with new material highlighted in red text.

Reviewer 2#

This study was designed to investigate the relationships between frailty and, health literacy (HL), including accessing, understanding, appraising, and applying health information, communication/interaction, and self-care behavior in patients with CKD. Authors found that management of patients with CKD should focus on young populations, reinforce health education strategies improving critical HL, and incorporate the fact that frailty may interfere with self-care. This manuscript gives new information, but several issues should be resolved to improve the manuscript.

Response to reviewer

Thank you for the comments. We will revise the manuscript according to your opinions.

  1. Authors adopted the CKDSC as a 16-item questionnaire with five subscales: medication adherence (five items), diet control (four items), exercise (three items), smoking behaviors (two items), and blood pressure monitoring (two items). Based on the Likert scale, responses ranged from 1 (almost never) to 5(almost always). Five items in the medication adherence subscale were negatively worded and needed to be reverse-recoded. Please explain and describe each item and make new tables for results of item as supplementary data.

Response to reviewer

Thank you for the comment. We have added the instrument of as supplementary data.

  1. Authors included patients aged 40 years or older. However, authors described the prevalence of CKD increases with age, especially in older adults (> 65 years) in the introduction part and titled as “Frailty, Health Literacy, and Self-Care in Elderly Adults with Chronic Kidney Disease in Taiwan”. Therefore, excluding patients aged < 65 years is more adequate for this manuscript. In addition, only 3 CKD 5 patients were enrolled in this study. Therefore, it is better to analyze data excluding CKD 5 patients.

Response to reviewer

Thank you for the comment.

  1. The aim of this study was to analyze the CKDSC and its’ related factors for CKD patients, not only for elderly. So it is not fit our purpose of the study if we excluding patients aged < 65 years. Moreover, after excluding the participants aged < 65 years, the small sample with only 88 subjects included for further statistic analysis. The generalization of the results could be limited.
  2. Although only 3 CKD 5 patients were enrolled in this study, we did not put each stage of CKD to regression model. We divided two groups for the stage of CKD as 3 stage and 4-5 stage. It is not violate the principle of statistical analysis.
  3. For the two reasons, we remain the 144 patients. We also concerned your suggestion, so the title has been revised avoiding misunderstanding the population in the study.

3.CKD-specific HL: The CKD-specific HL in Mandarin and Taiwanese was developed based on patient input, panel discussions with experts, and a literature review, and checked for validity and reliability in a pilot test. The factorial structure of the items was tested by confirmatory factor analysis, leaving 17 items to measure HL consisting of functional literacy (five items), communicative literacy (seven items), and critical literacy (five items). The 12 items were presented as a multiple-choice cloze test, with one point awarded for each correct response. Please explain and describe each item as supplementary data.

Response to reviewer

Thank you for the comment. We have added the instrument of as supplementary data.

Reviewer 3 Report

This is good research area but the manuscript needs signficant work.

Abstract

  • Modify the first line as "...often an irreversible condition..."
  • Don't use 'stage 3' - as this usage is now not recommended unless you clarify this to be based on GFR, in that case you may need to say G3. Please check the following for the latest nomenclature proposed by the KDIGO: https://www.kidney-international.org/article/S0085-2538(20)30233-7/fulltext
  • Good to present the time since diagnosis as mean with SD or median and range to provide better context.
  • I would assume the # 4 before conclusions is a typo?

Intro

  • Also here, use the KDIGO nomenclature, which has replaced the use of end-stage renal disease. Use kidney failure instead. 
  • The referencing on line 48 is a bit odd. Be consistent on your in-text citations.

Methods

  • On the use of the stage of CKD - please refer to my earlier comment.
  • More importantly, please provide detailed info on what definitions are used for the staging. What was the eGFR cut-off values?
  • Also change renal replacement therapy to kidney replacement therapy. Check latest evidence the earlier term is now almost obsolete.
  • Were there people in "stage 5" and were receiving supportive/palliative care than dialysis? If there were, did you exclude/include them?
  • You suddenly used Barthel Index on line 73, could be confusing for someone who don;t know what this index is. It is rather better to say those with poor activities of daily living/performance (Barthel Index<20)...
  • The analysis part is not detailed enough. For example, how dd you categorise them into low vs high self care? If there is a reference, please include it here as well. What regression? Logistic or linear?

Results

  • On Table 1, number of comorbid conditions is tricky. What qualifies to be included? Are these ongoing chronic conditions. And have you considered the use of Charlson's comorbidity index?
  • What is pre-frailty? Define this in the methods so that it is easy to understand the result.
  • I see you mentioned linear regression here. Please make sure these things are clearly included in the method. This is particulry important what variables were considered for the multivariate analysis and the criteria for their inclusion in the model.
  • You have the following in your discussion on line 167, please elaborate on this one to provide better context. Where were these studies from and what are major characteristics of patients targeted? "The proportion of patients with frailty in this study was higher than that in a study that adopted the same
    168 approach to diagnose frailty [29], but lower than that reported by Bilotta et al."
  • Same goes for the HL. It makes it easier to contextualise your study when you provide enough info re the studies you're comparing it against.
  • Re line 192-194, it is difficult to make the recommendation for inclusion of CKDSC as part of managment just based on this small sample cross-sectional study. It is better to recommend for further works (ideally those with follow-up) to understand if the CKDSC is a persistent issue that affect people across their illness trajectory and if it affects other patient outcomes. 

Author Response

Authors responses to decision letter

We are grateful for the positive and constructive comments that originated from the review process. We have responded to the points raised by the Editor and reviewers as follows, with new material highlighted in red text.

Reviewer 3# 

Abstract

1.Modify the first line as "...often an irreversible condition..."

  1. Don't use 'stage 3' - as this usage is now not recommended unless you clarify this to be based on GFR, in that case you may need to say G3. Please check the following for the latest nomenclature proposed by the KDIGO: https://www.kidney-international.org/article/S0085-2538(20)30233-7/fulltext
  2. Good to present the time since diagnosis as mean with SD or median and range to provide better context.
  3. I would assume the # 4 before conclusions is a typo?

Response to reviewer

  1. We have revised it. Please see the page 1, line 14.
  2. We have added the eGFR criteria of CKD, please see page 1, line 31-37.
  3. We have added the standard deviation to the mean of age and diagnosis time in abstract. Please see the page1, line 18-20.
  4. Thank you for the comment. We have deleted the error typing.

Introduction

  1. Also here, use the KDIGO nomenclature, which has replaced the use of end-stage renal disease. Use kidney failure
  2. The referencing on line 48 is a bit odd. Be consistent on your in-text citations.

Response to reviewer

  1. We have revised it and added the eGFR criteria of CKD, please see page 1, line 31-37.
  2. We have revised it. Please see the page 2, line 57.

Methods

  1. On the use of the stage of CKD - please refer to my earlier comment.
  2. More importantly, please provide detailed info on what definitions are used for the staging. What was the eGFR cut-off values?

Response to reviewer

We have added the eGFR criteria of CKD, please see page 1, line 31-37.

  1. Also change renal replacement therapy to kidney replacement therapy. Check latest evidence the earlier term is now almost obsolete.
  2. Were there people in "stage 5" and were receiving supportive/palliative care than dialysis? If there were, did you exclude/include them?
  3. You suddenly used Barthel Index on line 73, could be confusing for someone who don;t know what this index is. It is rather better to say those with poor activities of daily living/performance (Barthel Index<20)...
  4. The analysis part is not detailed enough. For example, how dd you categorise them into low vs high self care? If there is a reference, please include it here as well. What regression? Logistic or linear?

Response to reviewer

  1. We have revised renal replacement therapy to kidney replacement therapy. Please see the page 2, line 84.
  2. We have include the stage 5 who have not received the supportive/palliative care or dialysis.

10.We have revised it. Please see the page 2, line 85.

11 We adopted the multiple linear regression without categorizing self-care groups.

Results

  1. On Table 1, number of comorbid conditions is tricky. What qualifies to be included? Are these ongoing chronic conditions. And have you considered the use of Charlson's comorbidity index?
  2. What is pre-frailty? Define this in the methods so that it is easy to understand the result.
  3. I see you mentioned linear regression here. Please make sure these things are clearly included in the method. This is particulry important what variables were considered for the multivariate analysis and the criteria for their inclusion in the model.
  4. You have the following in your discussion on line 167, please elaborate on this one to provide better context. Where were these studies from and what are major characteristics of patients targeted? "The proportion of patients with frailty in this study was higher than that in a study that adopted the same
    168 approach to diagnose frailty [29], but lower than that reported by Bilotta et al."
  5. Same goes for the HL. It makes it easier to contextualise your study when you provide enough info re the studies you're comparing it against.
  6. Re line 192-194, it is difficult to make the recommendation for inclusion of CKDSC as part of managment just based on this small sample cross-sectional study. It is better to recommend for further works (ideally those with follow-up) to understand if the CKDSC is a persistent issue that affect people across their illness trajectory and if it affects other patient outcomes. 

Response to reviewer

12Thank you for the comment. On page 2, line 94, we calculate the numbers of these comorbid diseases. Although the Charlson's comorbidity index is a good measurement for comorbidity, the numbers of comorbid diseases did not a main issues for self-care of CKD. So we did not concern it.

  1. We have added the definition of pre-frailty on page 2-3, line 98-99.
  2. Thank you for the comment. We used the multiple linear regression not logistic regression and the error has been revised.

15 and16. We have added the description of related studies cited in our study to compare the results.

17 we have modified the suggestion. Please see the page 6, line 219-221.

Reviewer 4 Report

General comments

The authors have clearly stated that the purpose of the study was to investigate the relationships between frailty and health literacy, including accessing, understanding, appraising, and applying health information, communication/interaction, and self-care behavior in 59 patients with chronic kidney disease. The paper is well-written and easy to follow. In my opinion, it adds considerable value to the management of chronic kidney disease in older adults. Although the local focus of the present study, this work can enhance future attempts in similar research area conducted in a larger geographical scale. However, I have highlighted a few suggestions and concerns in my specific comments section (below) that need to be addressed before considering whether this work should be published or not. In summary, both introduction and discussion section need to be supported by additional evidence in order to present a robust article that provides readers with valuable remarks while showing the big picture.

Specific comments

  1. INTRODUCTION & DISCUSSION

- Nice work from the authors. I suggest adding a few sentences in the introduction aiming to connect the topic with the vital role of exercise in the management of CKD through the most attractive exercise types widely used in Asia and worldwide (1). Particularly, health/wellness coaching, exercise is medicine, fitness programs for older adults, and lifestyle medicine are some of the most popular health and fitness trends nowadays. This information should be linked to the present topic in some way in both introduction and discussion aiming to show the big picture, including the potential impact of exercise on functional health, frailty, health literacy, and well-being among patients with CKD. Additionally, a sentence regarding the rationale for exercise prescription as the magic pill by physicians (2) would be also be a useful addition based on the fact that regular exercise may be a critical part of the treatment puzzle in this cohort (3). Consider citing the following studies.

References:

  1. Kercher VM, Kercher K, Bennion T, Levy P, Alexander C, Amaral PC, et al. Fitness Trends from Around the Globe. ACSMs Health Fit J 2022; 26(1): 21-37.
  2. Sallis R, Franklin B, Joy L, Ross R, Sabgir D, Stone J. Strategies for promoting physical activity in clinical practice. Prog Cardiovasc Dis. 2015; 57(4): 375-386.
  3. Sovatzidis A, Chatzinikolaou A, Fatouros IG, Panagoutsos S, Draganidis D, Nikolaidou E, et al. Intradialytic Cardiovascular Exercise Training Alters Redox Status, Reduces Inflammation and Improves Physical Performance in Patients with Chronic Kidney Disease. Antioxidants 2020; 9(9): 868.

  1. RESULTS

If habitual physical activity (PA) were assessed, please add these data in Table 1 or 2. Otherwise, add one more limitation regarding these missing data given that PA is associated with general health.

  1. CONCLUSIONS

Please add more practical implications in this section.

Author Response

Authors responses to decision letter

We are grateful for the positive and constructive comments that originated from the review process. We have responded to the points raised by the Editor and reviewers as follows, with new material highlighted in red text.

Reviewer 4#

The authors have clearly stated that the purpose of the study was to investigate the relationships between frailty and health literacy, including accessing, understanding, appraising, and applying health information, communication/interaction, and self-care behavior in 59 patients with chronic kidney disease. The paper is well-written and easy to follow. In my opinion, it adds considerable value to the management of chronic kidney disease in older adults. Although the local focus of the present study, this work can enhance future attempts in similar research area conducted in a larger geographical scale. However, I have highlighted a few suggestions and concerns in my specific comments section (below) that need to be addressed before considering whether this work should be published or not. In summary, both introduction and discussion section need to be supported by additional evidence in order to present a robust article that provides readers with valuable remarks while showing the big picture.

Specific comments

  1. INTRODUCTION & DISCUSSION

- Nice work from the authors. I suggest adding a few sentences in the introduction aiming to connect the topic with the vital role of exercise in the management of CKD through the most attractive exercise types widely used in Asia and worldwide (1). Particularly, health/wellness coaching, exercise is medicine, fitness programs for older adults, and lifestyle medicine are some of the most popular health and fitness trends nowadays. This information should be linked to the present topic in some way in both introduction and discussion aiming to show the big picture, including the potential impact of exercise on functional health, frailty, health literacy, and well-being among patients with CKD. Additionally, a sentence regarding the rationale for exercise prescription as the magic pill by physicians (2) would be also be a useful addition based on the fact that regular exercise may be a critical part of the treatment puzzle in this cohort (3). Consider citing the following studies.

References:

  1. Kercher VM, Kercher K, Bennion T, Levy P, Alexander C, Amaral PC, et al. Fitness Trends from Around the Globe. ACSMs Health Fit J 2022; 26(1): 21-37.
  2. Sallis R, Franklin B, Joy L, Ross R, Sabgir D, Stone J. Strategies for promoting physical activity in clinical practice. Prog Cardiovasc Dis. 2015; 57(4): 375-386.
  3. Sovatzidis A, Chatzinikolaou A, Fatouros IG, Panagoutsos S, Draganidis D, Nikolaidou E, et al. Intradialytic Cardiovascular Exercise Training Alters Redox Status, Reduces Inflammation and Improves Physical Performance in Patients with Chronic Kidney Disease. Antioxidants 2020; 9(9): 868.

  1. RESULTS

If habitual physical activity (PA) were assessed, please add these data in Table 1 or 2. Otherwise, add one more limitation regarding these missing data given that PA is associated with general health.

  1. CONCLUSIONS

Please add more practical implications in this section.

Response to reviewer

Thank you for providing us the comments. The opinions made me confused  why reviewer suggested us to added the exercise related paper and the physical activity in our manuscript? Moreover, we did analyze if any factors were related to exercise. Why should we add the practical implication for exercise. Please give us more information for the comment.

Round 2

Reviewer 2 Report

The Authors answered every question. But minor issues should be resolved to improve the manuscript.

  1. Please add and describe supplementary data in the results section. In addition, explain about Mean±SD in the supplementary table of CKDSC (Medication adherence 4.8±0.6, 1. I myself may change the prescribed dosing time 4.7±0.7 2. I myself may stop taking drugs 4.8±0.5….)

Author Response

Authors responses to decision letter

We are grateful for the positive and constructive comments that originated from the review process. We have responded to the points raised by the Editor and reviewers as follows, with new material highlighted in red text.

Reviewer #2

The Authors answered every question. But minor issues should be resolved to improve the manuscript.

  1. Please add and describe supplementary data in the results section. In addition, explain about Mean±SD in the supplementary table of CKDSC (Medication adherence 4.8±0.6, 1. I myself may change the prescribed dosing time 4.7±0.7 2. I myself may stop taking drugs 4.8±0.5….)

Response to reviewer

We have explained the CKDSC in in the supplementary table. Please see the page 4, line 158-162.

Reviewer 3 Report

Most of my comments have been addressed, well done on that. But I strongly recommend you following the KDIGO definition & classification of CKD. Stage 3, 4 & 5 are not frequenly used now , it is rather G1 to G5 to indicate classification purely based on eGFR while A1 to A3 if there is any proteinuria involved etc. Closely look at this: https://www.sciencedirect.com/science/article/pii/S0085253815561747.

Please make this minor amendments when finalising your work.

Author Response

Authors responses to decision letter

We are grateful for the positive and constructive comments that originated from the review process. We have responded to the points raised by the Editor and reviewers as follows, with new material highlighted in red text.

#3

Most of my comments have been addressed, well done on that. But I strongly recommend you following the KDIGO definition & classification of CKD. Stage 3, 4 & 5 are not frequenly used now , it is rather G1 to G5 to indicate classification purely based on eGFR while A1 to A3 if there is any proteinuria involved etc. Closely look at this: https://www.sciencedirect.com/science/article/pii/S0085253815561747.

Please make this minor amendments when finalising your work.

Response to reviewer

We have revised the definition and classification of CKD in correspondence with KDIGO categories and added the related references. Please see the page 1, line 35-40.

Reviewer 4 Report

I have no additional comments. However, authors did not take into consideration my suggestions in the first round of revision regarding relevant references that can strengthen the introduction and discussion sections. Please consider citing the following studies.

  1. Kercher VM, Kercher K, Bennion T, Levy P, Alexander C, Amaral PC, et al. Fitness Trends from Around the Globe. ACSMs Health Fit J 2022; 26(1): 21-37.
  2. Sallis R, Franklin B, Joy L, Ross R, Sabgir D, Stone J. Strategies for promoting physical activity in clinical practice. Prog Cardiovasc Dis. 2015; 57(4): 375-386.
  3. Sovatzidis A, Chatzinikolaou A, Fatouros IG, Panagoutsos S, Draganidis D, Nikolaidou E, et al. Intradialytic Cardiovascular Exercise Training Alters Redox Status, Reduces Inflammation and Improves Physical Performance in Patients with Chronic Kidney Disease. Antioxidants 2020; 9(9): 868.

Author Response

Authors responses to decision letter

We are grateful for the positive and constructive comments that originated from the review process. We have responded to the points raised by the Editor and reviewers as follows, with new material highlighted in red text.

Reviewer #4

I have no additional comments. However, authors did not take into consideration my suggestions in the first round of revision regarding relevant references that can strengthen the introduction and discussion sections. Please consider citing the following studies.

  1. Kercher VM, Kercher K, Bennion T, Levy P, Alexander C, Amaral PC, et al. Fitness Trends from Around the Globe. ACSMs Health Fit J 2022; 26(1): 21-37.
  2. Sallis R, Franklin B, Joy L, Ross R, Sabgir D, Stone J. Strategies for promoting physical activity in clinical practice. Prog Cardiovasc Dis. 2015; 57(4): 375-386.
  3. Sovatzidis A, Chatzinikolaou A, Fatouros IG, Panagoutsos S, Draganidis D, Nikolaidou E, et al. Intradialytic Cardiovascular Exercise Training Alters Redox Status, Reduces Inflammation and Improves Physical Performance in Patients with Chronic Kidney Disease. Antioxidants 2020; 9(9): 868.

Response to reviewer #4

Thank you for the comment. We have listed one of your recommended reference in the study. Please see the page 5, line 186-187.